# Prognostic Value of Immune Environment Analysis in Small Bowel Adenocarcinomas with Verified Mutational Landscape and Predisposing Conditions

**DOI:** 10.3390/cancers12082018

**Published:** 2020-07-23

**Authors:** Erkki-Ville Wirta, Säde Szeto, Ulrika Hänninen, Maarit Ahtiainen, Jan Böhm, Jukka-Pekka Mecklin, Lauri A. Aaltonen, Toni T. Seppälä

**Affiliations:** 1Department of Gastroenterology and Alimentary Tract Surgery, Tampere University Hospital, 33520 Tampere, Finland; erkki-ville.wirta@fimnet.fi; 2Medical School, University of Helsinki, 00014 Helsinki, Finland; sade.szeto@helsinki.fi; 3Applied Tumor Genomics Research Program, Research Programs Unit, University of Helsinki, 00014 Helsinki, Finland; ulrika.hanninen@helsinki.fi (U.H.); lauri.aaltonen@helsinki.fi (L.A.A.); 4Department of Medical and Clinical Genetics, Medicum, University of Helsinki, 00014 Helsinki, Finland; 5Department of Education and Research, Central Finland Central Hospital, 40620 Jyväskylä, Finland; maarit.ahtiainen@ksshp.fi (M.A.); jukka-pekka.mecklin@ksshp.fi (J.-P.M.); 6Department of Pathology, Central Finland Central Hospital, 40620 Jyväskylä, Finland; jan.bohm@ksshp.fi; 7Faculty of Sport and Health Sciences, University of Jyväskylä, 40014 Jyväskylä, Finland; 8Department of Gastrointestinal Surgery, Abdominal Center, Helsinki University Hospital, 000290 Helsinki, Finland; 9Department of Surgery, University of Helsinki, 00014 Helsinki, Finland

**Keywords:** small bowel, adenocarcinoma, tumour infiltrating lymphocytes, PD-1, PD-L1

## Abstract

*Background:* Small bowel adenocarcinoma (SBA) is a rare yet insidious cancer with poor survival. The abundance of tumour-infiltrating lymphocytes is associated with improved survival, but the role of the programmed death-1/programmed death ligand-1 (PD-1/PD-L1) pathway in tumour escape is controversial. We evaluated immune cell infiltration, PD1/PD-L1 expression and their prognostic value in a series of SBAs with previously verified predisposing conditions and exome-wide somatic mutation characterization. *Methods:* Formalin-fixed paraffin-embedded tissue sections stained for CD3, CD8, PD-L1 and PD-1 were analysed from 94 SBAs. An immune cell score (ICS) was formed from the amount of the CD3 and CD8 positive lymphocytes from the tumour centre and invasive margin. The PD-L1 and PD-1 positive immune cells (ICs) and ICS were combined into a variable called Immunoprofile. *Results:* High ICS, PD-L1^IC^ and PD-1, individually and combined as Immunoprofile, were prognostic for better patient outcome. Sixty-five (69%) SBAs expressed ≥1% positive PD-L1^IC^. A high tumour mutation burden was common (19%) and associated with immune markers. Immunoprofile, adjusted for TNM stage, mismatch repair status, tumour location, sex and age were independent prognostic markers for disease-specific and overall survival. *Conclusions:* Analysing tumoral immune contexture provides prognostic information in SBA. Combining ICS, PD-1 and PD-L1^IC^ as Immunoprofile enhanced the prognostic performance.

## 1. Introduction

The small bowel is perhaps the most powerful organ in the human body in terms of regulating the immune system. This may explain why small bowel cancers are rare, comprising only approximately 3% of all GI-tract malignancies [1], although the incidence rate has been increasing [2]. Adenocarcinomas account for approximately 40% of all malignant small bowel tumours and are most commonly found in the duodenum (50–55%). Otherwise, 16–30% of tumours are found in the jejunum and 13–20% in the ileum [3,4]. Usually small bowel adenocarcinoma (SBA) occurs sporadically, but it is also often associated with several genetic predispositions, such as familial adenomatous polyposis (FAP), Lynch syndrome (LS), Peutz–Jeghers syndrome, and juvenile polyposis. Known risk factors for SBA also include Crohn’s disease and coeliac disease, which both cause chronic inflammation of the small bowel [5]. Environmental factors, like smoking, alcohol, red meat, sugar and starchy foods, have been reported to increase the risk of SBA [6]. In Crohn’s disease, the proposed additional risk factors for SBA include proximal disease of the intestine, male gender, diagnosis at a young age, surgically created non-functional bowel loops and the use of anti-inflammatory medications such as corticosteroids and tumour necrosis factor alpha antibodies [7].

Small bowel adenocarcinoma is often relatively asymptomatic for a long time, with the estimated period from first symptoms to diagnosis being 2–8 months. Therefore, SBA is discovered usually at an advanced stage and with a poor prognosis. In early stage disease, the reported 5-year overall survival ranges from 35% to 80%, in locally advanced disease with regional lymph node metastases 8% to 47% and with the disseminated disease with distant metastases only 5% [8]. Crohn’s disease associated SBA is usually located diffusely in the ileum or distal jejunum with similar symptoms as in active or obstructive Crohn’s disease, and it is rarely suspected preoperatively, resulting in an even worse outcome [9].

Most SBAs are believed to share a similar tumorigenesis with colorectal cancer (CRC), i.e., pathogenesis with a similar transition from adenoma to carcinoma [8]. However, chronic inflammation, as seen for example in Crohn’s disease, may cause SBA to arise through the inflammation–dysplasia–adenocarcinoma sequence [8,10,11]. The reason for the large difference between the incidence of CRC and SBA is not clear, but several possible explanations have been proposed. The small bowel epithelium has a faster renewal rate which prevents the accumulation of genetic damage, and the epithelial cells produce microsomal protective enzymes against food-derived carcinogens. The transit time of small bowel contents is faster, the large volume of secretions dilutes the concentration of carcinogens, and the low density of bacterial microbiota produce fewer carcinogenic metabolites, limiting strain on the small bowel mucosa. Gut-associated lymphoid tissue and various immune response-mediating cells, for example, cytotoxic CD8+ T-lymphocytes, are more prevalent in the small intestine, suggesting greater immune surveillance [12,13].

Analysis of the tumour microenvironment has offered a novel way to predict cancer behaviour. Antitumour immune surveillance is mainly overseen by CD8+ T-lymphocytes primed by specific dendritic cells located in the tertiary lymphoid structures. High intratumoral infiltration of CD3+ and CD8+ cytotoxic T-lymphocytes and adjacent tertiary lymphoid structures are associated with a favourable prognosis in patients with many different malignant tumours like melanoma, colorectal, non-small-cell lung, breast, head and neck, pancreatic and gastric cancers [14,15]. The Immunoscore, derived from the densities of CD3+ and CD8+ lymphocytes in the invasive margin and tumour centre, has proven to be a strong prognostic marker, especially in CRC, and could be used alongside traditional TNM-staging to better evaluate patient outcome [16].

Tumours with microsatellite instability (MSI) usually have a better prognosis than microsatellite stable (MSS) tumours, probably relating to the enhanced immune reaction against the massive mutation burden of the MSI tumours [17]. Although immune response is generally high in MSI tumours, Immunoscore has been shown to have prognostic meaning regardless of the tumour mismatch repair (MMR) status in colon cancer [18,19]. In SBA, the proportion of MSI tumours has been reported to vary between 5% and 35% [20], and, overall, the fraction of SBA with a high tumour mutational burden (TMB) seems to be greater than in CRC or gastric cancer [21].

One of the emerging hallmarks of cancer is avoiding host immune reaction by making the immune system unable to destroy cancer cells [22]. Evading immune surveillance can be done by inhibiting T-cell effector functions by upregulating the programmed cell death-1/programmed death ligand-1 (PD-1/PD-L1) pathway, which is seen in many cancers [22,23]. Normally, PD-1 is the surface receptor of an activated T-cell, and it mediates immunosuppression [24]. However, the exact role of the PD1/PD-L1 pathway in cancer progression is still insufficiently understood, and there is evidence of other tumour escape mechanisms, such as the local overproduction of prostaglandins that promote immunosuppressive and regulatory immune cells, allowing immune evasion [25,26]. In our earlier study on CRC, a high density of PD1- and PD-L1-positive immune cells were frequent in MSI tumours and associated with a strong immune reaction and favourable prognosis [25].

Small bowel adenocarcinoma is a rare and insufficiently understood cancer with a complex immune environment in the small bowel and several genetic or inflammatory predisposing diseases. Based on previous work, we aimed to evaluate immune contexture, including immune cell infiltration (Immune cell score, ICS) and PD1/PD-L1 expression, and their prognostic value in a nation-wide series of SBA with verified predisposing conditions and previous exome-wide somatic mutation characterization [27].

## 2. Results

### 2.1. Immune Contexture and Association with Other Clinicopathological Parameters

Immune cell score was ICS0 in 27 (29%), ICS1 in 14 (15%), ICS2 in 17 (18%), ICS3 in 11 (12%) and ICS4 in 24 (26%) tumours. A high number of PD-1 positive lymphocytes were expressed in 27 tumours (29%) including all the LS tumours. The PD-L1 positive immune cells (PD-L1^IC^) were expressed at >5% with moderate-to-strong staining intensity in 24 (26%) and PD-L1 positive tumour cells (PD-L1^TC^) in seven (7%) tumours. Although the >5% threshold was, here, selected for survival analysis, PD-L1^IC^ and PD-L1^TC^ with a ≥1% expression and moderate-to-strong staining intensity were observed in 65 (69%) and 14 (15%) tumours, respectively. Including tumours with weak staining intensity, a ≥1% expression of PD-L1^IC^ in 85% and PD-L1^TC^ was seen in 21% of the tumours. Immunoprofile was class 0 in 41 (44%), class 1 in 21 (22%), class 2 in 11 (12%) and class 3 in 19 (20%) tumours.

Immune cell score in association with other clinicopathological variables is shown in Table 1. Low ICS (0–2) and high ICS (3–4) did not associate statistically significantly with age, sex, tumour location, or coeliac disease. However, high ICS was associated with MSI (*p* = 0.005) and lower TNM stage (*p* = 0.022). Three of the four LS tumours had ICS4 and three of the four Crohn’s disease-associated tumours had IS0.

High PD-1 was associated with lower TNM stage so that only 9% of the stage IV tumours and 42% of the stage I–II tumours had high PD-1 (*p* = 0.005). In addition, 86% of the MSI tumours had high PD-1 (*p* < 0.001). Positive PD-L1^IC^ associated with MSI status (64% of the MSI tumours had positive PD-L1^IC^, *p* < 0.001). Immune cell score, PD-1 and PD-L1^IC^ were all strongly associated with each other (*p* < 0.001). Positive PD-L1^TC^ was only suggestively related to a higher tumour grade with 4/6 (67% with one case missing data) having tumour grade 3 (*p* = 0.034). High Immunoprofile was associated with lower TNM stage so that most of the stage IV tumours (88%) had a low Immunoprofile of 0 or 1 (*p* = 0.003). Immunoprofile was strongly associated with MSI status with 86% of the MSI tumours having Immunoprofile 2 or 3 (*p* < 0.001). All the LS tumours had a high Immunoprofile of 2 or 3 (*p* = 0.018, Table 2).

The associations of TMB with clinicopathological variables are shown in Table 3. High TMB was strongly associated with MSI tumours, high ICS, high PD-L1^IC^ and PD-1 expression and high Immunoprofile. Four (5%) of the MSS tumours had high TMB. High TMB was also associated with coeliac disease with five (56%) tumours expressing high TMB (*p* = 0.012, Table 3).

No significant associations were found between the *KRAS*, *APC*, or *TP53* mutations and ICS, PD-L1^IC^, PD-1, or Immunoprofile. Correlations between different mutation types in the coding regions and mutational signatures and immune markers are shown in Appendix A for MSS tumours and in Appendix A for MSI tumours. Only mutational signature 17 had a weak correlation with PD-L1^IC^ (*r*_s_ = 0.318, *p* = 0.004) and Immunoprofile (*r*_s_ = 0.269, *p* = 0.017) in MSS tumours.

### 2.2. Univariable Survival Analyses

Kaplan–Meier survival analyses with clinicopathological variables are shown in Appendix A. High ICS, high PD-L1^IC^, high PD-1 and high Immunoprofile (Figure 1) were significantly prognostic for better patient outcome (5-year disease-specific survival (DSS) of 70% versus 30%, 81% versus 33%, 80% versus 31% and 93% versus 26% and 5-year overall survival (OS) of 68% versus 25%, 77% versus 28%, 74% versus 27% and 93% versus 21%, respectively, *p* < 0.001 for all). Median OS for low ICS, low PD-L1^IC^, low PD-1 and low Immunoprofile were 27, 31, 31 and 27 months, respectively, and median DSS was 31 months for all. For high ICS, high PD-L1^IC^, high PD-1 and high Immunoprofile median OS or DSS were not reached. Similarly, patients with high TMB had an improved DSS (5-year survival of 39% versus 77%, *p* = 0.009) and OS (5-year survival of 33% versus 72%, *p* = 0.003). Median DSS for low TMB was 41 and median OS 33 months, for high TMB median DSS or OS were not reached. A high TNM stage was prognostic for poor survival, with stage IV having only a 7% 5-year DSS and OS (*p* < 0.001 for both, median DSS 15 and median OS 13 months). Tumour MSI status was prognostic for better DSS (5-year survival of 39% versus 85%, *p* = 0.003) and OS (5-year survival of 34% versus 79%, *p* = 0.001). Median DSS for MSS tumours was 41 and median OS 34 months, for MSI tumours median DSS or OS were not reached. Tumour location in the duodenum or jejunum rather than in the ileum had a better DSS (5-year survival of 50% and 52% versus 34%, *p* = 0.045) and OS (5-year survival of 47% and 51% versus 21%, *p* = 0.014). Median DSS for duodenal tumours was not reached, for jejunal tumours 68 months and for ileal tumours 22 months. Median OS for duodenal, jejunal and ileal tumours were 48, 55 and 21 months, respectively. Furthermore, patients with coeliac disease had a favourable prognosis for DSS (5-year survival of 78% versus 42%, *p* = 0.045) and OS (5-year survival of 78% versus 37%, *p* = 0.014). Median DSS or OS for coeliac disease were not reached. For other than coeliac disease associated tumours, median DSS was 45 and median OS 36 months.

### 2.3. Multivariable Survival Analysis

According to the univariable analysis, TNM Stage, MMR status, ICS, PD-1, PD-L1^IC^ and tumour location were included in the multivariable analysis together with age and sex. Coeliac disease was excluded because of the small sample size, especially after the exclusion of tumours with inadequate data.

Table 4 shows the multivariable model with separate immune parameters, ICS, PD-1 and PD-L1^IC^. Of the immune parameters, only PD-L1^IC^ was found to be an independent prognostic marker with DSS, with a hazard ratio (HR) of 4.73 (*p* = 0.020) and an OS HR of 3.88 (*p* = 0.018). Stage IV, with respect to stage I as a reference, was highly prognostic for poor DSS and OS (HR 17.16, *p* = 0.001 and HR 6.64, *p* < 0.001). Tumour location in the ileum was prognostic for worse patient outcome (HR 8.25 for DSS, *p* < 0.001 and HR 5.76 for OS, *p* = 0.001). In addition, MSS tumours had a worse OS (12.18, *p* = 0.025).

Table 5 shows the multivariable model with Immunoprofile adjusted for age, gender, TNM stage, MMR status and tumour location. Immunoprofile was dichotomized into low (classes 0 and 1) and high (classes 2 and 3) groups, because after the exclusion of the tumours with inadequate data, the original groups sizes were small. Furthermore, according to the Kaplan–Meier analysis, the DSS and OS among low and high Immunoprofile classes were at a similar level (Appendix A). The prognostic effect of Immunoprofile was enhanced compared to the effect of separate immune parameters. For DSS, low Immunoprofile had an HR of 6.34 (*p* = 0.008) and for OS HR of 3.57 (*p* = 0.022). The TNM stage remained as an independent prognostic factor (stage IV HR 32.91 for DSS and 13.24 for OS, *p* < 0001 for both) as well as tumour location in the ileum (HR 3.37 for DSS, *p* = 0.001 and HR 2.83 for OS, *p* = 0.002). Age > 60 was prognostic for worse DSS (HR 2.44, *p* = 0.038) and OS (HR 2.61, *p* = 0.018).

Of the 14 MSI tumours, four were excluded because of insufficient data. Among those were two of the three MSI patients who succumbed during follow-up including one MSI tumour with an SBA related death and low Immunoprofile. Mismatch repair status was not prognostic in a model with Immunoprofile, and all but one MSI tumour left in the model had a high Immunoprofile. In a multivariable model with separate immune parameters but without MMR status, PD-L1^IC^ remained the only independent prognostic factor (HR 5.87, *p* = 0.009 for DSS and HR 4.94, *p* = 0.007 for OS, Appendix A). In a model with Immunoprofile and with the MMR status removed, the effect of Immunoprofile was even more pronounced, with low Immunoprofile having an HR of 11.96 for DSS and an HR of 7.54 for OS, *p* < 0.001 for both (Appendix A). Impact of coeliac and Crohn’s disease were evaluated in an additional multivariable model but both were left insignificant (Appendix A).

## 3. Discussion

Our results indicate that analysing tumour immune contexture may be helpful in predicting tumour behaviour in SBA. Immune cell infiltration and high PD-1 and PD-L1 expression in tumour-infiltrating lymphocytes were common in MSI tumours and were associated with a lower TNM stage. All the immune parameters were highly prognostic in the univariable survival analysis, yet only PD-L1^IC^ remained an independently significant single marker in the multivariable model for DSS and OS in the presence of MMR status. High ICS, PD-1 positivity and PD-L1 expression in tumour-infiltrating lymphocytes all had strong relationships with each other, causing a possible confounding effect when evaluated together. However, Immunoprofile, determined as a combination of ICS, PD-1 and PD-L1^IC^, enhanced the effect of single variables and proved to be a strong independent prognostic factor for DSS and OS as we have previously reported in CRC [25].

An earlier study of the SBA immune environment reported a high expression of PD-1 in peritumoral and intratumoral lymphocytes (83%) and PD-L1 in both tumour cells (17%) and tumour-infiltrating lymphocytes (43%) [28]. We used a cut-off value of >5% for PD-L1 positivity with moderate-to-strong staining intensity, which was more informative in the survival analysis and provided comparative results with earlier reports in our previous study performed on CRC [25]. However, with the same cut-off value of 1% used by Thota et al. [28] for PD-L1 positivity, our sensitivity analyses showed positive PD-L1TC in 15% and positive PD-L1IC in 69% of the tumours with at least moderate staining intensity. When including tumours with weak staining intensity, a ≥1% expression of PD-L1TC was seen in 21% and PD-L1IC in 85% of the tumours. Our study population had a significantly higher share of stage IV tumours (37% versus 20%) than the SBA cohort presented by Thota et al. [28]. Of the stage IV tumours, 42% expressed ≥1% positive PD-L1IC, and 20% expressed ≥1% positive PD-L1TC with moderate-to-strong staining intensity. As the PD-1/PD-L1 pathway provides a possible immune escape route for tumours, it is expected that PD-L1 expression in advanced disease is increased.

Although PD-1/PD-L1 expression should suppress the immune reaction against tumours, the reports of the prognostic value of PD-L1 expression vary among different cancer types. The expression of PD-L1 has been shown to correlate with a poor prognosis in oesophageal cancer, pancreatic carcinoma, hepatocellular carcinoma, renal cell carcinoma and ovarian cancer [29,30,31,32,33,34]. However, PD-L1 correlated with better patient outcome in breast cancer and Merkel cell carcinoma [35,36]. For lung cancer, melanoma, gastric cancer and CRC both positive and negative prediction values have been reported [37,38,39,40,41,42,43,44,45].

One possible reason for this discrepancy between results may be explained by the large variation of definitions of PD-L1 positivity, with the cut-off for positive staining varying from 1% to 50%. In some studies, staining was considered positive only with moderate-to-strong staining intensity, while others considered also weak staining intensity as positive. In addition, some have evaluated only the overall positivity of the tumour sample, while others have identified positive tumour cells and positive immune cells separately [46,47]. It has also been reported that PD-L1 not only inhibits T-cell proliferation and cytokine production, but also enhances T-cell activation, and the explanation for this is still unknown [48]. Considering all the contradictory information, it is clear that the function of the PD-1/PD-L1 pathway is not yet fully understood. In our study, the obvious explanation for the favourable effect of positive PD-L1IC and PD-1 would be the close association of PD-1/PD-L1 expression with a large amount of tumour-infiltrating lymphocytes, so that PD-1/PD-L1 expression rather reflects enhanced host immune reaction against tumour cells, which is balanced by tumour-derived immune suppression.

Microsatellite unstable tumours are characterized by an extensive mutational load, leading to truncating mutations that are identified by the immune surveillance because of misfolded proteins serving as neoantigens, causing enhanced antitumoral immune reaction and, therefore, improved survival. This was noted also in our results, as MSI status was associated with high ICS, a high number of PD-1 positive immune cells and PD-L1IC expression. The incidence of MSI tumours was 15%, which is in accordance with earlier reports [20]. Response to immune checkpoint inhibitors correlates with TMB, MMR status and PD-L1 expression. There is evidence that even very low (1%) PD-L1IC positivity may be sufficient to predict the efficiency of PD-L1 blockade, and that the adverse effects of the treatments seem to be higher with PD-L1 negative patients [49]. In a large study by Salem et al. [50], 4125 tumours from 14 different cancer sites, including 147 SBAs, were quantified for TMB, MMR status and PD-L1 expression. They found that SBA had one of the highest TMB rates (14% with a cut-off of 17 mutations/Mb), and TMB was strongly associated with MSI status [50]. In our study, we selected a cut-off of 10 mutations/Mb, which in a trial of nivolumab plus ipilimumab in non-small-cell lung cancer-selected patients was most likely to have a response irrespective of tumour PD-L1 expression level [51]. As expected, high TMB was associated with MSI and a better Immunoprofile with high ICS, PD-L1IC and PD-1. High TMB was discovered in 18 (19%) SBAs including all the 14 MSI tumours. Every MSI tumour had at least 22.84 mutations/Mb, qualifying as having a high TMB status also with several large mutational landscape studies [50,52,53]. As the MSI status is relatively frequent, the TMB rates are high and PD-L1IC positivity is common, selected SBAs could potentially be good responders for checkpoint inhibitor therapy.

Despite many similarities, SBAs have a distinct mutational landscape compared to CRC or gastric cancer [27]. We found that the only notable correlation between different mutation patterns and immune parameters was the correlation of mutational signature 17 with PD-L1IC, and, therefore, with Immunoprofile in MSS tumours. Signature 17 has been found also in both CRC and gastric cancers, but the meaning and the process causing it is still unclear [27,54]. High ICS was associated with high TMB, but we found no correlations between high immune cell infiltration and specific mutation types. Some tumours with high TMB had a low ICS and Immunoprofile. However, all three Immunoprofile 0 tumours with a high TMB were at an advanced TNM stage (III or IV), indicating successful cancer immune evasion.

It is known that coeliac disease increases the risk of SBA, especially in refractory coeliac disease where mucosal damage remains even with a strict gluten-free diet [55]. However, the prognosis of coeliac disease associated with SBA seems to be significantly improved compared to SBA without coeliac disease [56,57]. This was explained by the high frequency of MSI tumours (73%) and overall lower TNM stage upon discovery [56]. In our study, coeliac-associated tumours showed better survival in the univariable analysis, but the number of tumours was too small to be included in further analysis. The proportion of MSI tumours was 44% (4/9), and only one of the tumours was at stage IV. Four of the patients had an Immunoprofile of 0 (three stage III and one stage IV tumour), including the only two coeliac disease patients with SBA-related death during follow-up. None of the tumours expressed high PD-L1TC, which presented a trend for worse DSS in the univariable analysis. It is also possible that improved survival is related to the enhanced clinical follow-up of coeliac disease patients, i.e., SBA is diagnosed at an earlier stage.

Tumour location in the ileum was prognostic for worse survival for DSS and OS, resulting in the high portion of stage IV tumours in the ileum. Most of the coeliac-associated tumours were located in the jejunum (67%), which is in accordance with previous studies [56,57]. The overall distribution of SBAs in our study differs from large epidemiological studies, where the duodenum is the most common location [3,4]. The higher incidence of duodenal tumours may be related to the abundant exposure to bile [58]. Here, 53% of the tumours were located in the jejunum and only 18% in the duodenum, arguably reflecting our small cohort size. However, exclusion of the tumours of the papillary region may have contributed to the difference.

A weakness of this study is the deficiency of the clinical data, and therefore less than a quarter of the study population were excluded from the multivariable analysis. In addition, because the tumour type is rare, the study population was small to begin with, and the study is retrospective, the results can be considered only as suggestive. However, we provide new information on SBA, as we have analysed survival data of SBAs with a previously characterized mutational profile in the context of the immune landscape with ICS, PD-1 and PD-L1IC positivity determined from whole-section FFPE samples. Until now, Immunoprofile [25] has lacked validation in an independent dataset that we now present herein, suggesting that Immunoprofile is a scalable predictor of cancer outcome across tumour types.

## 4. Material and Methods

### 4.1. Patients

The tumours used in the present study were derived from a nation-wide series of 106 SBAs treated in Finnish hospitals between 2003 and 2011, described previously [27]. In 94 cases, representative tumour samples were still available for immunohistochemical analyses. Tumours of the papillary region were excluded since they may have originated in the pancreas or the biliary tract. The median age of the patients during surgery was 62 (interquartile range 54–73) and there was a slight female predominance (53%). The median follow-up time after surgery was 44 months (interquartile range 16–84 months). Nine tumours (10%) were associated with coeliac disease and four (4%) with Crohn’s disease. Four patients (4%) had LS and two (2%) had FAP. Fourteen tumours (15%) were MSI. The tumour location was the duodenum in 18%, the jejunum in 53%, the ileum in 19% and undetermined in 10% of cases. Tumours with coeliac disease were mainly located in the jejunum (67%) and tumours with Crohn’s disease were found equally in the jejunum and ileum. Stage distribution was I 4%, II 21%, III 26%, IV 37% and unknown in 12%. Three patients died post-operatively, and three patients had insufficient survival data and were excluded from the survival analysis. Ten mutations per megabase (Mb) was selected as a cut-off to determine high TMB.

### 4.2. Immunohistochemical Analyses

Formalin-fixed paraffin-embedded (FFPE) tissue sections of 3 µm thickness were used for the immunohistochemical analyses. Staining for PD-1 and PD-L1 was conducted with anti-PD-1 (SP269, 1:50; Spring Bioscience) and anti-PD-L1 (E1L3N, 1:100; Cell Signaling Technology) antibodies, using a BOND-III stainer (Leica Biosystems). Staining for CD3 and CD8 was conducted with anti-CD3 (LN 10, 1:200; Novocastra) and anti-CD8 (SP16, 1:400; Thermo Scientific) antibodies using a Lab Vision Autostainer 480 (ImmunoVision Technologies Inc.). Signal visualization was done by diaminobenzidine, and sections were counterstained with haematoxylin. The slides were scanned with a NanoZoomer-XR (Hamamatsu Photonics) at 20× magnification (Figure 2).

### 4.3. Scoring

Scoring was done as described earlier [25]. Briefly, representative areas from the tumour centre and the invasion margin were selected, and positively stained CD8, CD3 and PD-1 lymphocytes were calculated by using QuPath [59]. The invasive margin was selected manually using an annotation brush with a diameter of 720 µm. The analyses were done by two examiners and interrater reliability (IRR) was evaluated by determining the intraclass correlation coefficient. Inter-rater reliabilities for the analyses of CD3, CD8 and PD-1 tumour centre were 0.96 (CI95% 0.93–0.97), 0.98 (CI95% 0.96–0.99) and 0.73 (CI95% 0.57–0.83), respectively. Inter-rater reliabilities for analyses of the CD3, CD8 and PD-1 invasive margin were 0.89 (CI95% 0.83–0.92), 0.90 (CI95% 0.84–0.94) and 0.75 (CI95% 0.58–0.84), respectively. The mean analysed areas for the tumour centre from CD3-, CD8-, and PD-1-stained sections were 24.9 mm², 22.3 mm^2^ and 18.5 mm², and for the invasion margin 12.2 mm², 11.8 mm^2^ and 10.5 mm², respectively. Cut-off values for ICS were selected from receiver operating characteristic curves (715 for CD3+, 255 for CD8+, and 48 for PD-1+ in the tumour centre; 1327 for CD3+, 598 for CD8+, and 27 for PD-1+ in the invasive margin). Cut-off values were the same for 5 and 10 year disease-specific survival related ROC curves. Patients were divided into low ICS (scores 0–2) and high ICS (scores 3–4) groups for further analysis. Expression of PD-1 was considered positive if the cell count was high on both the tumour centre and invasive margin.

Expression of PD-L1 was evaluated on tumour cells (TCs) and tumour-infiltrating immune cells (ICs) throughout the tumour centre and the invasive margin. Both the percentage of stained immune and tumour cells and the staining intensity were visually estimated. Tumour samples were defined as PD-L1 positive when >5% of the tumour cells and/or tumour-infiltrating immune cells were positive for PD-L1 with moderate or strong intensity [25].

As established previously, ICS, PD-1 cell count and PD-L1 expression in tumour infiltrating lymphocytes were combined into a single immune parameter called Immunoprofile [25]. High ICS, high PD-1 and positive PD-L1^IC^ each accounted for 1 point. As an example, when ICS and PD-1 were high and PD-L1^IC^ was positive, the Immunoprofile was 3, and when ICS and PD-1 were low and PD-L1^IC^ negative, the Immunoprofile was 0 [25]. Immunoprofile was further categorised as low (0–1) or high (2–3). Immune cell score and PD-1 were indeterminable from one tumour, as was PD-L1. The Immunoprofile was therefore missing for two tumours.

### 4.4. Statistical Analysis

Categorical data were compared using Pearson’s chi-square or Mantel–Haenszel tests. Fisher’s exact test was used for variables with a low observation rate. The Spearman correlation coefficient (*r*_s_) was used to determine correlations between mutational data and immune variables. The Kaplan–Meier method was used to calculate disease-specific survival (DSS) and overall survival (OS), and the log-rank test was used to compare differences. A *p*-value < 0.05 was considered as a threshold of statistical significance. Survival times for DSS and OS were calculated from the date of surgery to the time of death or the end of follow-up. Death within 30 days following surgery was considered post-operative. Univariable and multivariable Cox proportional hazards regression models were used to analyse prognostic factors for DSS and OS. Only variables with a *p*-value < 0.20 in the univariable analysis were included in the multivariable analysis with age and gender. Statistical analyses were performed using IBM SPSS Statistics (version 25.0; SPSS Inc., Chicago, IL, USA).

### 4.5. Ethical Aspects

The study was reviewed and approved by the Ethics Committee of the Hospital District of Helsinki and Uusimaa, Finland (408/13/03/03/2009). Authorization for the use of the patient registry and genetic studies was obtained from the National Supervisory Authority for Welfare and Health (Valvira). Informed patient consent was waived by the Ethics Committee.

## 5. Conclusions

In conclusion, we found that a high portion of SBAs express ≥1% of PD-L1 in immune cells within the tumour environment, and a high TMB is relatively common, although mainly seen in MSI tumours. Therefore, SBA tumours might be good responders to immune checkpoint therapy. High ICS, PD-L1^IC^, PD-1 and Immunoprofile were associated with a high TMB. The only correlation between mutation types and signatures and immune parameters was the weak correlation of mutational signature 17 and PD-L1^IC^ in MSS tumours. However, the true significance of this finding is unclear and requires further research.

Immune cell score, PD-1 and PD-L1^IC^ were prognostic for DSS and OS in the univariable analysis, but only PD-L1^IC^ remained so in the multivariable analysis. Still, the high interaction among immune markers may cause a confounding effect. By combining the immune markers into a single variable called Immunoprofile, the prognostic effect was enhanced. Immunoprofile, adjusted for TNM stage, MMR status, tumour location, sex and age, was an independent prognostic marker for DSS and OS in SBA.

## Figures and Tables

**Figure 1 cancers-12-02018-f001:**
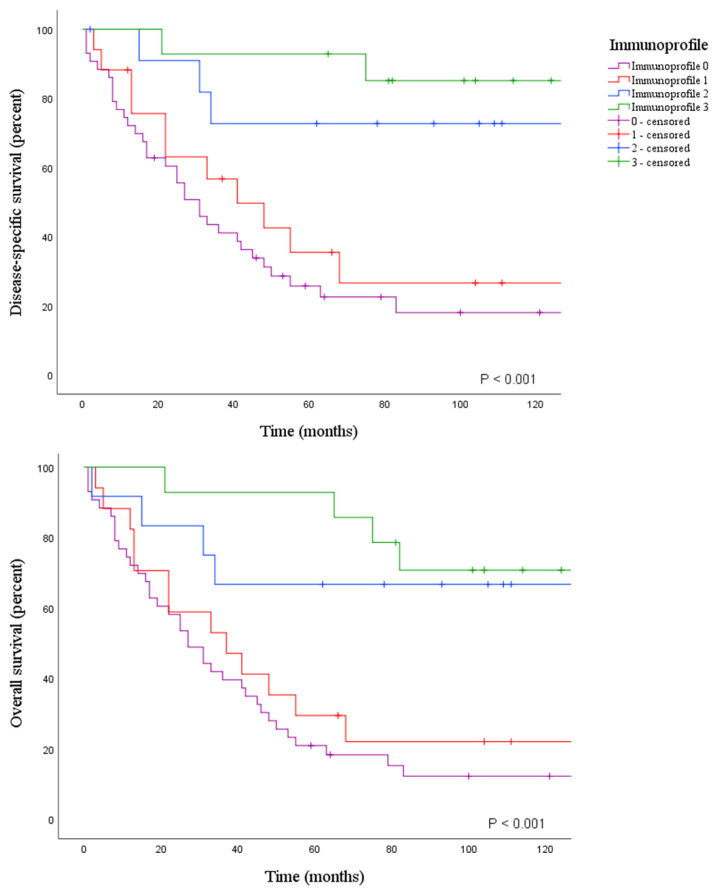
Disease-specific and overall survivals according to Immunoprofile.

**Figure 2 cancers-12-02018-f002:**
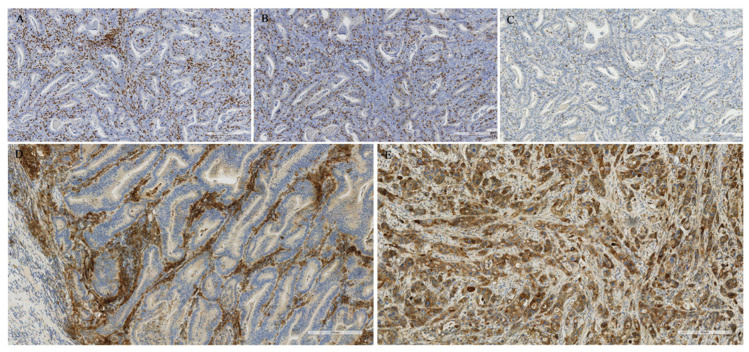
Examples of CD3 (**A**), CD8 (**B**) and PD-1 (**C**) stainings. (**D**) PD-L1 positive staining in immune cells and (**E**) PD-L1 positive staining in tumour cells. Images were captured with 8× zoom from 20× magnified scanned slides.

**Table 1 cancers-12-02018-t001:** Clinicopathological variables and their associations with Immune cell score.

	Low ICS	High ICS	Total	*p*-Value
	*n* = 58	*n* = 35	*n* = 93	
	(% of Row)	(% of Row)	(% of the Column)	
**Age**				
<60 years	23 (61)	15 (40)	38 (41)	0.761
≥60 years	35 (64)	20 (36)	55 (59)	
**Sex**				
Female	29 (66)	15 (34)	44 (47)	0.504
Male	29 (59)	20 (41)	49 (53)	
**TNM stage**				
I	1 (25)	3 (75)	4 (4)	0.022 *
II	11 (55)	9 (45)	20 (22)	
III	12 (50)	12 (50)	24 (26)	
IV	26 (76)	8 (24)	34 (37)	
**Tumour grade**				
1	10 (59)	7 (41)	17 (19)	0.889 *
2	35 (63)	21 (38)	56 (63)	
3	9 (56)	7 (44)	16 (18)	
**MMR status**				
MSI	4 (29)	10 (71)	14 (15)	0.005
MSS	54 (68)	25 (32)	79 (85)	
**Tumour location**				
Duodenum	9 (53)	8 (47)	17 (18)	0.292
Jejunum	33 (66)	17 (34)	50 (54)	
Ileum	9 (53)	8 (47)	17 (18)	
**Hereditary syndromes**				
Lynch syndrome	1 (25)	3 (75)	4 (4)	0.265
FAP	1 (50)	1 (50)	2 (2)	
No	56 (64)	31 (36)	87 (94)	
**Crohn’s disease**				
Yes	3 (75)	1 (25)	4 (4)	1.000 **
No	55 (62)	34 (38)	89 (95)	
**Coeliac disease**				
Yes	4 (44)	5 (56)	9 (10)	0.289 **
No	54 (64)	30 (36)	84 (90)	

* Mantel–Haenszel test was used; ** Fisher’s exact test was used; TNM stage in eleven, grade in four and location in nine tumours was unknown. ICS was undeterminable from one tumour. Abbreviations: ICS, immune cell score; TNM, tumour-nodes-metastasis; MMR, mismatch repair; MSS, microsatellite stable; MSI, microsatellite instability; FAP, Familial adenomatous polyposis.

**Table 2 cancers-12-02018-t002:** Association of clinicopathological variables to Immunoprofile.

	Immunoprofile 0	Immunoprofile 1	Immunoprofile 2	Immunoprofile 3	Total	*p*-Value
	*n* = 47	*n* = 18	*n* = 13	*n* = 14	*n* = 92	
	(% of Row)	(% of Row)	(% of Row)	(% of Row)	(% of Column)	
**Age**						
<60 years	18 (49)	8 (22)	5 (14)	6 (16)	37 (40)	0.967
≥60 years	29 (53)	10 (18)	8 (15)	8 (15)	55 (60)	
**Sex**						
Female	23 (54)	9 (21)	5 (12)	6 (14)	43 (47)	0.894
Male	24 (49)	9 (18)	8 (16)	8 (16)	49 (53)	
**TNM stage**						
I	1 (25)	1 (25)	0 (0)	2 (50)	4 (5)	0.003 *
II	8 (40)	3 (15)	4 (20)	5 (25)	20 (25)	
III	8 (35)	5 (22)	6 (26)	4 (17)	23 (28)	
IV	22 (65)	8 (24)	2 (6)	2 (6)	34 (42)	
**Tumour grade**						
1	8 (47)	4 (24)	4 (24)	1 (6)	17 (19)	0.544 *
2	29 (53)	9 (16)	8 (15)	9 (16)	55 (63)	
3	7 (44)	4 (25)	1 (6)	4 (25)	16 (18)	
**MMR status**						
MSS	45 (58)	18 (23)	8 (10)	7 (9)	78 (85)	<0.001
MSI	2 (14)	0 (0)	5 (36)	7 (50)	14 (15)	
**Location**						
Duodenum	7 (41)	3 (18)	4 (24)	3 (18)	17 (21)	0.563
Jejunum	27 (55)	9 (18)	6 (12)	7 (14)	49 (59)	
Ileum	7 (41)	5 (35)	1 (6)	3 (18)	17 (21)	
**Lynch syndrome**						
no	47 (53)	18 (21)	11 (13)	12 (14)	88 (96)	0.018
yes	0 (0)	0 (0)	2 (50)	2 (50)	4 (4)	
**Crohn’s disease**						
no	45 (51)	17 (19)	13 (15)	13 (15)	88 (96)	0.821
yes	2 (50)	1 (25)	0 (0)	1 (25)	4 (4)	
**Coeliac disease**						
no	43 (52)	17 (21)	11 (13)	12 (15)	83 (90)	0.745
yes	4 (44)	1 (11)	2 (22)	2 (22)	9 (10)	

* Mantel–Haenszel test was used; TNM stage in eleven, grade in four and location in nine tumours was unknown. Immunoprofile was undeterminable from two tumours.

**Table 3 cancers-12-02018-t003:** Association of tumour mutational burden with clinicopathological variables.

	TMB-Low	TMB-High	Total	*p*-Value
	(% of Row)	(% of Row)	(% of Column)	
**Age**				
<60 years	32 (82)	7 (18)	39 (41)	0.803
≥60 years	44 (80)	11 (10)	55 (59)	
**Sex**				
Female	36 (82)	8 (18)	44 (47)	0.823
Male	40 (80)	10 (20)	50 (53)	
**pTNM stage**				
I	3 (75)	1 (25)	4 (5)	0.674 *
II	16 (80)	4 (20)	20 (24)	
III	18 (75)	6 (25)	24 (29)	
IV	29 (83)	6 (17)	35 (42)	
**Tumour grade**				
1	15 (88)	2 (12)	17 (19)	0.166 *
2	46 (81)	11 (19)	57 (63)	
3	11 (69)	5 (31)	16 (18)	
**MMR**				
MSS	76 (95)	4 (5)	80 (85)	<0.001
MSI	0 (0)	14 (100)	14 (15)	
**ICS**				
Low	52 (90)	6 (10)	58 (62)	0.005
High	23 (66)	12 (34)	35 (38)	
**PD-L1^IC^**				
Low	61 (88)	8 (12)	69 (74)	0.001
High	14 (58)	10 (42)	24 (26)	
**PD-L1^TC^**				
Low	70 (81)	16 (19)	86 (92)	0.617
High	5 (71)	2 (29)	7 (8)	
**PD-1**				
Low	60 (91)	6 (9)	66 (71)	<0.001
High	15 (56)	12 (44)	27 (29)	
**Immunoprofile**				
0	44 (94)	3 (6)	47 (51)	<0.001 *
1	15 (83)	3 (17)	18 (20)	
2	8 (62)	5 (39)	13 (14)	
3	7 (50)	7 (50)	14 (15)	
**Tumour location**				
Duodenum	15 (88)	2 (12)	17 (20)	0.738
Ileum	40 (80)	10 (20)	50 (59)	
Jejunum	15 (83)	3 (17)	18 (21)	
**Coeliac disease**				
No	72 (85)	13 (15)	85 (90)	0.012 **
Yes	4 (44)	5 (56)	9 (10)	
**Crohn’s disease**				
No	72 (80)	18 (20)	90 (96)	1.000 **
Yes	4 (100)	0 (0)	4 (4)	

Abbreviations: TMB, tumour mutational burden; ICS, immune cell score; PD-L1, programmed death ligand-1; IC, immune cell; TC, tumour cell; PD-1, programmed cell death protein-1.* Mantel–Haenszel test was used; ** 2-sided Fisher’s Exact test was used. TNM stage in eleven, grade in four and location in nine tumours was unknown. ICS and PD-1 were indeterminable from one tumour, as was PD-L1. The Immunoprofile was therefore missing for two tumours.

**Table 4 cancers-12-02018-t004:** Multivariable analysis with Immune cell score, PD-1 and PD-L1^IC^.

	Univariable Analysis	Disease-Specific Survival		Overall Survival	
		(*n* = 68)		(*n* = 68)	
	P	HR (95% CI)	*p*-Value	HR (95% CI)	*p*-Value
**Age**					
<60 years	DSS: 0.614 OS: 0.640	1	0.146	1	0.052
≥60 years		1.89 (0.80–4.47)		2.22 (0.99–4.96)	
**Sex**					
Female	DSS: 0.829 OS: 0.536	1	0.251	1	0.446
Male		0.65 (0.31–1.35)		0.76 (0.38–1.54)	
**TNM stage**					
I	DSS: <0.001 OS: <0.001	1	0.001	1	<0.001
II		2.03 (0.18–22.44)		0.62 (0.11–3.42)	
III		3.15 (0.30–32.67)		0.85 (0.16–4.52)	
IV		17.16 (1.44–204.47)		6.64 (1.08–41.07)	
**MMR status**					
MSS	DSS: 0.010 OS: 0.004	5.83 (0.66–51.41)	0.112	12.18 (1.37–108.21)	0.025
MSI		1		1	
**Immune cell score**					
low	DSS: 0.001 OS: <0.001	1.96 (0.79–4.85)	0.145	2.33 (0.98–5.57)	0.056
high		1		1	
**PD-1**					
low	DSS: <0.001 OS: <0.001	1.13 (0.27–4.78)	0.868	0.51 (0.14–1.84)	0.307
high		1		1	
**PD-L1^IC^**					
low	DSS: <0.001 OS: <0.001	4.73 (1.27–17.57)	0.020	3.88 (1.26–11.93)	0.018
high		1		1	
**Tumour location**					
Duodenum	DSS: 0.043 OS: 0.015	1	<0.001	1	<0.001
Jejunum		0.88 (0.25–3.08)		1.00 (0.31–3.21)	
Ileum		7.83 (1.55–39.62)		7.26 (1.57–33.54)	

Abbreviations: HR, hazard ratio; CI, confidence interval; DSS, disease-specific survival; OS, overall survival; MMR, mismatch repair; MSS, microsatellite stable; MSI, microsatellite instable; PD-1, programmed cell death protein 1; PD-L1, programmed death ligand 1; IC, immune cell. Analyses were performed with the following reference categories: <60 years, female gender, TNM Stage I, MSI status, high Immune cell score, high PD-1, high PD-L1IC and duodenal tumour location. For analyses there were 68 patients available. Eleven patients were excluded because of unknown TNM stage, one patient had insufficient samples for Immune cell score and PD-1, one patient had insufficient samples for PD-L1IC, nine patients had unknown tumour location, three patients had insufficient survival data, and two patients were excluded because of post-operative death.

**Table 5 cancers-12-02018-t005:** Multivariable analysis with Immunoprofile.

	Univariable Analysis	Disease-Specific Survival		Overall Survival	
		(*n* = 68)		(*n* = 68)	
	P	HR (95% CI)	*p*-Value	HR (95% CI)	*p*-Value
**Age**					
<60 years	DSS: 0.614 OS: 0.640	1	0.038	1	0.018
≥60 years		2.44 (1.05–5.67)		2.61 (1.18–5.79)	
**Sex**					
Female	DSS: 0.829 OS: 0.536	1	0.053	1	0.091
Male		0.49 (0.24–1.01)		0.55 (0.28–1.10)	
**TNM stage**					
I	DSS: <0.001 OS: <0.001	1	<0.001	1	<0.001
II		2.79 (0.27–28.68)		1.03 (0.20–5.36)	
III		6.26 (0.66–59.29)		2.06 (0.45–9.42)	
IV		32.91 (2.88–375.66)		13.24 (2.23–78.55)	
**MMR status**					
MSS	DSS: 0.010 OS: 0.004	5.23 (0.57–47.57)	0.142	8.90 (1.00–79.39)	0.050
MSI		1		1	
**Immunoprofile**					
Low	DSS: <0.001 OS: <0.001	6.34 (1.61–24.97)	0.008	3.57 (1.20–10.62)	0.022
High		1		1	
**Tumour location**					
Duodenum	DSS: 0.043 OS: 0.015	1	0.001	1	0.002
Jejunum		0.58 (0.18–1.90)		0.57 (0.19–1.74)	
Ileum		3.37 (0.79–14.32)		2.83 (0.70–11.41)	

Abbreviations: HR, hazard ratio; CI, confidence interval; DSS, disease-specific survival; OS, overall survival; MMR, mismatch repair; MSS, microsatellite stable; MSI, microsatellite instable. Analyses were performed with the following reference categories: <60 years, female gender, TNM Stage I, MSI status, high Immunoprofile and duodenal tumour location. For analyses there were 68 patients available. Eleven patients were excluded because of unknown TNM stage, two patients were excluded because of insufficient samples for Immunoprofile, nine patients had unknown tumour location, three patients had insufficient survival data and two patients were excluded because of postoperative death.

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
