# Peer review of "Prognostic Value of Immune Environment Analysis in Small Bowel Adenocarcinomas with Verified Mutational Landscape and Predisposing Conditions"

_cancers, 2020, doi:10.3390/cancers12082018_

Round 1
Reviewer 1 Report
This is a well-written paper by a research group that has applied immune cell quantification analysis in multiple GI cancers in the past. They collected an impressive series of relatively rare cancer, small bowel adenocarcinoma, and performed an analysis of Immunoprofile. Immunoprofile consists of CD3, CD8, PD1 and PD-L1 cell counts at tumor invasive margins and tumor core, and in combination inform of the status of the tumor microenvironment. By utilizing this method previously introduced in colorectal cancer, they also validated the applicability of the approach now with a another type of GI disease. The method adds value to the widely spread Immunoscore developed by Galon et al in France. Although this is a retrospective analysis, the results are interesting in addition to the literature and to what is known about SBA. The results are based on methods and the conclusions are based on results. The limitations of the study are thoroughly presented. Minor suggestions - Please consider adding Supplementary figure 1 to the main text as an example of the staining quality. This would also strengthen the discussion about the PD-L1 staining.Author Response
Minor suggestions - Please consider adding Supplementary figure 1 to the main text as an example of the staining quality. This would also strengthen the discussion about the PD-L1 staining.
Answer: We agree that supplementary figure 1. is better suited in the main text and it is now relocated to r354–57 as suggested.
Reviewer 2 Report
In the present study Wirta et al showed that CD3/CD8 lymphocyte infiltration and the pattern of PD1/PD-L1 expression may predict the prognosis and survival outcome in a series of 94 patients with small bowel adenocarcinoma.
This is an interesting paper, addressing a poorly investigated issue. Methods are adequate and results have been reported in a comprehensive manner. I have only few minor objections.
1) In table 4, it would be better to add celiac disease and Crohn's disease in multivariate analysis to assess whether such diseases may influence survival analysis.
2) Paragraph 2.2: please report medians of survival as well.
Author Response
1) In table 4, it would be better to add celiac disease and Crohn's disease in multivariate analysis to assess whether such diseases may influence survival analysis.
Answer: We have added a new multivariable model with coeliac disease and Crohn’s disease to the supplementary appendix as a Supplementary Table S5, and the results has been stated on r209-10. We agree that inclusion of celiac and Crohn’s disease would be interesting addition for multivariable analysis. Unfortunately, only four patients had Crohn’s disease associated tumour and only eight patients had coeliac disease associated tumour (one was excluded from the multivariable model because of unknown tumour location) so statistical impact cannot be reliably assessed. Univariable analysis of overall survival for Crohn’s disease had P-value of 0.300, which is not sufficient for inclusion in the multivariable model according to our methods, which is why we suggest it to be included in the supplements. Both coeliac and Crohn’s disease were left insignificant in the new multivariable analysis.
2) Paragraph 2.2: please report medians of survival as well.
Answer: We have added median survival times to the paragraph 2.2.